# DECO-Bench: Unified Benchmark for Decoupled Task-Agnostic Synthetic Data Release

**Farzaneh Askari**[*,1,2], **Lingjuan Lyu**[1], **Vivek Sharma**[*,†,1]

[1]Sony AI, Sony Research,
[2]McGill University

farzaneh.askari@mail.mcgill.ca, lingjuan.lv@sony.com, viveksharma@sony.com

## Abstract

In this work, we tackle the question of how to systematically benchmark task-agnostic decoupling methods for privacy-preserving machine learning (ML). Sharing datasets that include sensitive information often triggers privacy concerns, necessitating robust decoupling methods to separate sensitive and non-sensitive attributes. Despite the development of numerous decoupling techniques, a standard benchmark for systematically comparing these methods remains absent. Our framework integrates various decoupling techniques along with synthetic data generation and evaluation protocols within a unified system. Using our framework, we benchmark various decoupling techniques and evaluate their privacy-utility trade-offs. Finally, we release our source code, pre-trained models, datasets of decoupled representations to foster research in this area.

## 1 Introduction

The advancement of machine learning and its integration into real-world applications are rapidly accelerating due to globalization and the abundance of data. However, the use of all data sources remains limited because of compliance with modern privacy standards. The privacy-preserving machine learning community has addressed this issue by proposing decoupling techniques for raw data. These methods aim to alleviate privacy concerns by separating sensitive from non-sensitive information and anonymizing the sensitive data.

Imagine a party owning a large dataset of face images with well-annotated attributes (e.g., race, age, gender, emotions). This data could benefit many research studies and applications, such as face recognition and age prediction. However, privacy concerns hinder the sharing of this data. The data holder cannot foresee all potential use cases before releasing the dataset, making it essential to employ task-agnostic decoupling methods as a privacy-preserving measure. The representation decoupling studies often utilize adversarial representation learning [32, 52, 17, 44, 31, 37, 38, 45, 35, 47, 48], including non-linear neural networks and non-convex optimization. Consequently, they lack formal privacy guarantees (in contrast with differential privacy, [14, 15]) and their privacy and utility performance is evaluated empirically and varies across studies.

Our goal is to propose a benchmarking framework to systematically evaluate decoupling techniques. By benchmarking these techniques, we aim to accelerate privacy-preserving research and create a level playing field for researchers. While our framework is designed to be task and data agnostic, this study focuses on visual data, specifically image datasets and image classification for benchmarking and dataset release.

---

[*]Equal contribution. [2] This work was conducted while FA was doing internship at Sony AI. [†]VS started and led the project. Correspondence to: Vivek Sharma.

38th Conference on Neural Information Processing Systems (NeurIPS 2024) Track on Datasets and Benchmarks.

Our approach includes several steps. First, we elaborate on current decoupling techniques and their implementation. Second, we discuss synthetic image generation with decoupled attributes and the practical aspects of this process. Third, we introduce metrics and evaluation protocols for comparing the performance of decoupling methods. Finally, we conduct extensive experiments using our framework and provide a detailed discussion on the use of our benchmarking framework. We want to emphasize that our motivation for this benchmark is to provide a systematic evaluation of various decoupling techniques, especially when offering a theoretical guarantee is not feasible. We meticulously design various aspects of our experiments and highlight the significance of each component while benchmarking decoupling techniques. The summary of our contribution is as follows:

**1. Framework:** We develop a comprehensive framework that integrates various decoupling techniques, a synthetic image generation pipeline, and standardized evaluation metrics. Fig. 1 demonstrates various components of our framework. **2. Decoupling and Synthesizing Integration:** we integrate the decoupling methods, in the context of synthetic image generation, and discuss the practical considerations around them. **3. Dataset:** We have released a dataset of images with decoupled attributes, generated using our image generation pipeline. This dataset enables researchers to evaluate decoupling methods within our framework, utilizing the provided metrics. **4. Benchmark:** We establish a comprehensive benchmark by extensively evaluating several decoupling techniques for privacy-preserving image classification. This benchmark systematically quantifies the privacy-utility trade-off and highlights the impact of practical design decisions on privacy while evaluating decoupling techniques.

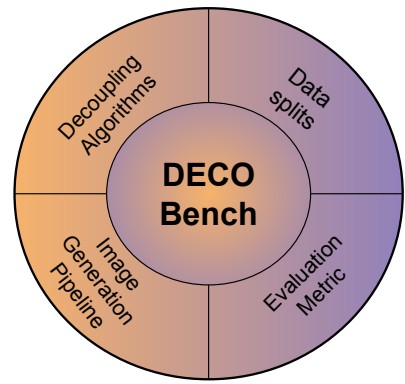

Figure 1: Architecture of DECO-Bench. Our framework has four main components. i) Decoupling algorithms, ii) Image generation pipeline, iii) Structured data splits, and iv) Evaluation metrics for utility and privacy. Our framework provides the users with all the key modules to train and evaluate their decoupling technique. For more information, see Sec. 2 and 3.

## 2 Decoupling Algorithms

In this section, we review the adversarial representation-learning-based decoupling algorithms integrated in our framework, including their formulas and the underlying intuition. Consider a party holding dataset $D^{private} = \{X, A\}$ with N pair of samples $(x, a)$ where $x \in X$ and $a \in A$. For example, $x$ represents an image of a person's face with attributes such as age, gender, race, emotions, etc. denoted as $a$. The data holder party views certain attributes as sensitive (e.g., race) and others as non-sensitive (e.g., age); denoted as $a_S$ and $a_{NS}$ respective. The attributes $a_S$ and $a_{NS}$ are mutually exclusive, such that $a = \{a_S \cup a_{NS}\}$.

In order to protect the sensitive attributes, the data holder party refrains from releasing their dataset. This caution is especially warranted because releasing only the non-sensitive attributes (i.e., $x, a_{NS}$) may still allow for the retrieval of sensitive attributes through existing correlations in the data samples. For example, having the original image of a person's face, one can readily infer their race, age, etc. Therefor, the problem of protecting sensitive attributes in not trivial.

The goal of attribute decoupling techniques is to acquire a decoupled version of $D^{private}$ (noted as $D^{deco}$) with the same distribution and N pair of samples, $(x', a')$ where the sensitive information is anonymized (i.e. **privacy**). While it is crucial for $D^{deco}$ to protect private information, it is equally important for it to maximally preserve the usability of dataset in other tasks such as reconstructions and the use of non-sensitive attributes. In attribute decoupling, this is referred to as **utility**. Given that sensitive and non-sensitive attributes might be correlated, there is often a privacy-utility trade-off using decoupling techniques.

Assume that a model $f_\theta(x) = z$ is trained with the objective of learning the latents to maximize a utility task. The decoupling methods aim to learn a model that separates the input $x$ into sensitive and

non-sensitive latents, denoted as $z_S$ and, $z_{NS}$ respectively ($z_S, z_{NS} \in z$). Assuming that an adversary will attempt to infer $a_S$ from $z_S$, the goal is to decouple the $z_S$ latent space so that reconstructing the sensitive attributes is not possible. The decoupling methods, introduce a proxy adversary during training (with gradient propagation) that attempts to recover the sensitive attributes $a_S$ from $z$. Then an implicit *min-max* optimization takes place, where the main objective's loss is minimized (utility) while the adversary's loss is maximized (privacy). Intuitively, the goal is to restructure the latent space, such that $a_S$ is not retrievable. Below, we review some of the decoupling methods.

**Cross-Entropy Decoupling (CE):** Consider an adversary model that attempt in recognizing sensitive attributes from the latent space in the form of a classification task: $f_\phi^a(z) = a_S$. Given the model and the proxy adversary, CE decoupling minimizes the utility loss while maximizing the adversary classification loss. The overall objective can be summarized as $min_{\theta,\phi} L_{util}(\theta, \phi) + max_{\theta,\phi} L_{priv,CE}(\theta, \phi)$. As a result, during the training, the latent space is arranged in such a way that the adversary's ability to infer $y_S$ is restricted [47].

**Cross-Entropy and masking Decoupling (CE + Mask):** In addition to maximizing a cross entropy loss for the proxy adversary, some studies [47] include a dynamic Filter Generating Network (FGN) [25, 46]. FGN is a learnable mechanism, trainable end-to-end with the base and adversary model, that generates binary masks to zero out the feature map channels contributing to attribute leakage while preserving those crucial to the utility task.

**Metric Learning:** With the same principal as CE decoupling method, the decoupling algorithms using metric learning [11] aims to decouple the latent space, so that sensitive attributes are not retrievable from the latents. In supervised metric learning [12, 20], the goal is to learn a distance metric and organize the latent space such that the latent representations for examples in the same class (positive pairs) are brought closer together, while those in different classes (negative pairs) are pushed further apart. To anonymize the representation space with respect to the sensitive attribute $a_S$, a metric loss is maximized so that the positive pairs are distanced while the negative pairs are brought closer together.

The methods we described above are the underlying methods that are used in decoupling or obfuscation studies. Although, DECO-Bench is inspired by them; none of these studies address the decoupling problem in synthetic image generation setup. Therefore, one of our contribution is generating synthetic images while preserving privacy using the decoupling techniques. In Sec. 4.2 we elaborate on how we integrate these techniques in synthetic image generation setup and discuss the practical implication.

# 3 Framework Setup

In this section, we employ image classification and datasets of face images as a use-case to detail our framework. We elaborate on its various components and their practical considerations.

**Base Model:** our base model in this benchmark is based on Latent Diffusion Model (LDM) [22, 42]; which means, the base training objective is to denoise and reconstruct the input by predicting the noise added during the forward diffusion process. The input to an LDM is the noisy representation of the image in the latent space of pretrained autoencoders. Therefore, during training, the objective is to minimize the mean squared error loss between the original and the predicted noise ($f_\theta$). To integrate the decoupling technique, we add the mechanisms discussed in Sec. 2 ($f_\phi$) to the main objective.

**Image Generation:** LDM is a powerful tool for synthetic image generation, conditioned on an initial input such as image prompt, text prompt or both. Using the trained base model parameters, we generate a synthetic dataset, conditioned on the original private dataset $D^{private}$. This synthetic dataset includes images with sensitive and non-sensitive attributes decoupled, serving as $D^{deco}$.

**Data Splits:** An important and often overlooked practical consideration in benchmarking is the data splits. In our framework, we define clear splits of datasets and ensure the equal distributions of attributes among the sets using multi-label stratification [49]. Specifically, we define four main data portions: $D^{train}$ ($Tr$), $D^{val}$ ($V$), $D^{gen}$ ($G$) (gen, short for generation), and $D^{test}$ ($Test$). For the remaining of the paper, when we mention $G$, we refer to the synthetic version; unless otherwise mentioned (i.e., $G_{real}$).

Using $Tr$ we train the base and adversary model and validate it on $V$. We then generate $G$ from $G_{real}$ using our trained LDM. This decoupled dataset is used to train the utility and privacy evaluation classifiers. Finally, the classifiers are tested on $Test$. Creating distinct splits of data, allows for fair and structured evaluation of methods.

**Evaluation and Metrics:** The decoupling method results in a privacy-utility trade-off. While it is important to decouple the attributes and anonymize the sensitive attributes; it is equally important to maintain the utility quality. Therefore, the evaluation metrics need to address both privacy and utility performance.

− **Utility FID Evaluation:** We measure the utility performance by assessing the quality of generated images in $G$. Fréchet Inception Distance (FID) [21] is a popular metric for measuring the quality of generated images created by generative models. In our case, we use FID to compare the distribution similarity between $G$ and $G_{real}$.

− **Utility CLS Evaluation:** Another metric to measure the utility performance is to quantify the classification of non-sensitive attributes from synthetic images. To measure the reconstruction of non-sensitive information from generated decoupled data; we train a classifier on the sample pairs $(x', a'_{NS})$ from $G$ and evaluate it on the unseen test portion of $Test$. We can say the utility is preserved if the classifier has a high classification accuracy.

− **Privacy CLS Evaluation:** Different from the traditional definition of privacy in differential privacy literature [14, 15]; we measure the privacy by the ability to recognize the sensitive attribute from $G$. To do this, we train a classifier to classify sensitive attributes from generated images. Ideally, the classification accuracy should be around random, indicating that it is not possible to recognize the sensitive attribute from the generated dataset.

− **Privacy Pretrained (PT) Evaluation:** we propose a second criterion to evaluate privacy performance. Using the trained base and adversary model, we run inference on $G_{real}$ (before synthesizing) and measure classification accuracy. This pretrained evaluation protocol quantifies privacy performance in isolation from the effect of image generation pipeline and serves as a complementary metric to privacy CLS evaluation.

# 4 Experiments and Results

In this section, we first introduce the datasets and the design decisions we applied. We then elaborate on training and generation pipelines and our experiment details. Finally, we conclude the section by sharing the quantitative and qualitative results and discussing our findings.

## 4.1 Datasets:

In our setup, we do not collect real world data and instead propose our benchmark using public image classification datasets. We specifically use the FairFace [29] and UTKFace [56], datasets of face images with annotated attributes. For both of these datasets, we used cropped and aligned versions, ensuring the person's face is positioned in the center of the image.

**FairFace:** [29] is a dataset of face images created to address the ethnicity imbalance present in other face datasets. The dataset includes 108,501 images of people's faces, annotated with age, gender, and race, comprising nine, two, and seven classes, respectively. In our decoupling experiments, we use race as the sensitive attribute, while gender and age are treated as non-sensitive attributes.

**UTKFace:** [56] is another popular face datasets containing 20,000 face images and includes annotations for person ethnicity, race, and gender. We use race as a sensitive attribute and gender and age as utility attributes.

**Cross Label Mapping:** To facilitate cross-evaluation across the datasets, we defined common sets of labels for both datasets. Specifically, we mapped the UTKFace age labels to match the FairFace age labels. Similarly, we grouped the race labels in FairFace to correspond with the UTKFace race labels. More specifically, for FairFace race labels, we grouped "East Asian" and "Southeast Asian" into one label corresponding to the "Asian" label in UTKFace, and grouped "Latino_Hispanic" and "Middle Eastern" together to represent the "Other" category in UTKFace. Likewise, we mapped the

Table 1: Dataset Statistics for FairFace and UTKFace datasets. The number of classes for the datasets after cross label mapping are demonstrated (see Sec. 4.1).

| Dataset | Train (Tr) Samples | Val (V) Samples | Gen (G) Samples | Test Samples | Privacy Attr. (#Classes) | Utility Attr. (#Classes) | Eval Metric |
|---------|--------------------|-----------------|-----------------|--------------|--------------------------|--------------------------|-------------|
| FairFace | 15614 | 1735 | 6953 (691 val) | 3833 | Race (C=5) | Age (C=9), Gender (C=2) | Accuracy |
| UTKFace | 12694 | 1410 | 6044 (604 val) | 3566 | Race (C=5) | Age (C=9), Gender (C=2) | Accuracy |

high quality images          low quality images

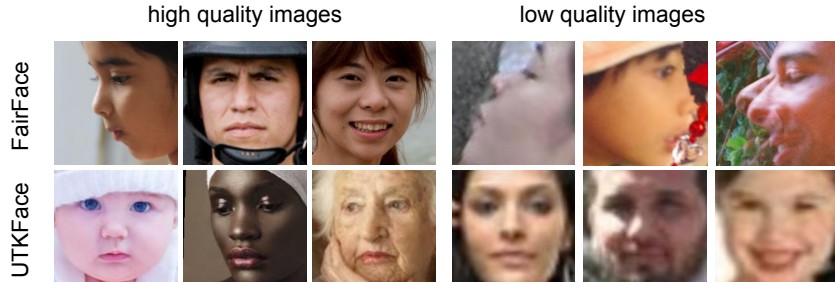

Figure 2: High and low quality samples in FairFace and UTKFace datasets.

age annotations from UTKFace (*i.e.*, 0-116) to match the FairFace age annotations (*i.e.*, 0-2, 3-9, and so on).

**Dataset Statistics:** As discussed earlier, a key component of our framework, is the definition and structuring of data splits to report results clearly, keeping the dataset size in mind. Specially, since the CLS evaluation protocol requires training a classifier, the size of data plays an important role and should not be overlooked. In our setup for each dataset, we allocate 0.60, 0.25, and 0.15 of dataset for $Tr$, $G$, and $Test$ respectively. From, $Tr$ we reserve 0.1 of the split as validation set ($V$). Likewise, we reserve 0.1 of the $G$ split for validation. It is crucial to note that each dataset is split using its original label set (without grouping) and based on multi-label stratification [49]. To ensure corresponding splits include a similar number of examples, we use only a portion of the FairFace dataset. Tab. 1 summarizes the datasets statistics. Fig. 2 shows a few examples from both datasets with high quality and low quality images.

## 4.2 Experiment Setup

**Training:** For our base model, we train a Stable Diffusion (SD) v1-5 [42, 1] with conditional Unet backbone model [43] using Low Rank Adaption (LoRA) [23]. Using the SD model with conditional Unet backbone, we add the LoRA adaptors to the last convolutional projection layer of every cross-attention block in the backbone (*i.e.*, downsample blocks, mid blocks, and upsample blocks). The idea is to have the adaptors right at the output of each transformer-based cross attention layer. We apply these layers in a balanced manner across the down, mid, and upsampling blocks. In total, we add 4.5% to 4.7% the total number of model parameters as LoRA adapters and finetune the model on our datasets. During training in addition to finetuning the model on training split of our dataset in hand, our goal is to restructure the learned latent space, such that the sensitive attribute is anonymized. To achieve this, we average the output of these LoRA layers (LoRA features) to handle varying resolution across cross attention layers of conditional Unet and follow the decoupling methods.

The images from $Tr$ are first mapped to the latent space using a pretrained variational autoencoder [28] (frozen during training). The latents are diffused with noise using a DDPM noise scheduler [22]. The output of this model is the prediction of the initial noise that was added to the latents. In addition to noisy latents, the model also receives a text prompt, as a condition. Using all the attributes in the dataset, we form a simple prompt in the format of "A {Age} year-old {Race} {Gender}". A pretrained CLIP tokenizer and encoder [39] is used for text encoding (frozen during training). We train the base model using the objectives discussed in Sec. 2. We benefit from an LR scheduler and early stopping criterion and save the LoRA weights before the model overfit on the $V$ split. It is important to note that the base SD model is pretrained on laion-aesthetics dataset [30].

**Decoupling Methods:** Although the decoupling methods discussed in Sec. 2 are existing studies; to the best of our knowledge, none is used in the context of synthetic image generation. In this section, we elaborate on the integration of each method in our pipeline. **CE:** during the training, we extract

the LoRA features. Given the block from different depth of Unet backbone have various spatial size, we apply an adaptive average pool on LoRA features. The average pooled representations are input to a Fully Connected layer (FC) for classification of sensitive attribute, followed by a CE loss. During training, we maximize this classification CE loss while minimizing the diffusion MSE loss. The intuition behind this method is obfuscating the latent space such that the sensitive attribute cannot be discriminated using the latent representations. **CE+Mask**: this method is built upon the CE, with an additional FGN [47, 46] to apply masking on the output of the first LoRA adapter. We add a trainable FC layer right after the first LoRA adapter, that outputs a feature map score. These feature map scores are weakly discretized (using temperature sigmoid) and thresholded to output a binary mask. Finally the binary mask is applied to the output of the first LoRA layer $(L_1)$ which is passed down to the rest of the model. The masked out $L_1$ is also input to the adversary classification along with the rest of LoRA features. The ratio of masked channels is a hyperparameter. The idea of CE+mask is to apply a data driven channel pruning method such that the sensitive information is selectively obfuscated. **Metric Learning:** supervised metric learning aims to learn a metric such that the samples with same labels are brought closer together and pushed apart from samples with different labels. However, we seek the opposite effect; meaning, we aim to bring different labels of sensitive attributes (e.g., ethnicity) closer together in the latent space such that the latent space cannot be used to discriminate between them. Therefore, we use a semi-hard negative and positive miner [20, 36] on the average pooled LoRA features to retrieve the negative pairs and positive pairs. We make a triplet where the anchor remains the same, while we swap negative and positive pairs. These triplets are input to a triple margin loss which is minimized as usual, with the only difference that positive and negative pairs are swapped. This will apply the desired effect of restructuring the latent space so that the sensitive attribute (anchor) is not easily recognizable. In all the methods discussed above, once the training is complete using each of these methods, we save the LoRA weights and load them in the generation pipeline. For each layer that we created a LoRA adapter, we completely replace the weights with LoRA weights.

**Synthetic Image Generation:** In order to generate the decoupled dataset, we use images in $G_{real}$ as initial images along with a prompt using the sensitive and non-sensitive attributes. Using the pretrained model and the LoRA weights (with LoRA weights fused to the base model), we generate synthetic images from $G_{real}$. Each image sample from the $G_{real}$ set along with its text prompt (similar to prompts used during training) is input to the generation pipeline. The result is a synthetic dataset, $(G)$. Given the training and decoupling process, ideally, the images in $G$ should not represent the same sensitive attribute (in our setup, race) as the original image while retaining the non-sensitive attributes according to the original prompt image. The synthetic images are saved for the remaining of the pipeline.

**Training and Generation Setups:** Since our goal is to offer a benchmark to comprehensively evaluate various aspects of decoupling methods, we design three setups to train and evaluate our models. **Setup A:** Our first setup is using different non-overlapping splits of the FairFace dataset to train $(Tr)$, validate $(V)$, generate $(G)$ from $(G_{real})$, and evaluate $(Test)$. **Setup B:** In this setup B, we evaluate the generalization abilities of the model across datasets; where the base model is trained and validated on $Tr$, $V$ of Fairface dataset and $G$ is generated and evaluated on $G_{real}$ and $Test$ splits of UTKFace dataset. **Setup C:** provides a comparison for both setup A and B, by training the base model jointly on the $Tr$ splits of FairFace and UTKFace. The trained model is used to generate $G$ from $G_{real}$ split of each dataset and be used for evaluating on $Test$ split of the respective dataset. It is important to note both $G_{real}$ and $Tr$ have a held out validation sets to be used during training. During the generation, the validation set of $G_{real}$ remains untouched, to match the data distribution of $Test$.

**Privacy, Utility Evaluation Setup:** Finally, we train the utility and privacy classifiers on the $G$ dataset and test it on the $Test$. For PT evaluation (see Sec. 3), we take the trained model on $Tr$ set and evaluate it on $G_{real}$. For utility and privacy CLS evaluation, we train a ResNet18 [18] classifier on the $G$ dataset for each setup and report the classification accuracy on the $Test$. We use learning-rate scheduler along with early stopping based on validation loss and save the best model to report the performance. It is important to note that the classification labels are cross mapped across the two datasets.

**Other Experiments:** Given that quantification of the privacy-utility performance of decoupling method, involves an extra step of training a classifier; the size of $G_{real}$ and in turn $G$ plays a crucial role in reporting performance. Therefore, we design a set of experiments to emphasize the importance

Table 2: **Setup A**: Trained and Tested on the Fairface dataset. Classification accuracy percentage (CLS and PT) for privacy leakage and utility on synthetic data.

| | Method | Privacy (PT, Race, ↓) | Privacy (CLS,Race,↓) | Util (CLS,Age,↑) | Util (CLS,Gender, ↑) |
|---|---|---|---|---|---|
| **G** | Baseline (Real) | − | 50.12 | 36.60 | 72.94 |
| | CE | 19.27 | 41.64 | 33.81 | 70.47 |
| | CE + Mask | 14.18 | 37.07 | 34.44 | 68.77 |
| | Metric Learning | − | 45.40 | 34.20 | 69.42 |
| **V+G** | Baseline (Real) | − | 55.86 | 39.45 | 74.69 |
| | CE | 19.22 | 46.23 | 34.44 | 73.13 |
| | CE + Mask | 14.59 | 38.22 | 35.30 | 73.99 |
| | Metric Learning | − | 37.49 | 28.72 | 75.16 |

Table 3: **Setup B**: Trained Fairface and Tested on UTKFace dataset. Cross generalization classification accuracy percentage for privacy leakage and utility.

| | Method | Privacy (PT, Race, ↓) | Privacy (CLS,Race,↓) | Util (CLS,Age,↑) | Util (CLS,Gender, ↑) |
|---|---|---|---|---|---|
| **G** | Baseline (Real) | − | 72.30 | 49.47 | 86.19 |
| | CE | 42.52 | 64.26 | 41.31 | 84.36 |
| | CE + Mask | 16.87 | 63.72 | 43.87 | 83.77 |
| | Metric Learning | − | 63.10 | 41.03 | 85.07 |
| **V+G** | Baseline (Real) | − | 75.79 | 51.1 | 86.7 |
| | CE | 42.50 | 66.62 | 42.46 | 84.96 |
| | CE + Mask | 16.70 | 65.61 | 43.76 | 84.84 |
| | Metric Learning | − | 64.15 | 40.52 | 83.69 |

of data splits size on the reported performance of decoupling methods. **V+G:** In this setup, we generate synthetic images on the $V$ data split of each dataset and add it to the $G$. We then carry out the rest of the pipeline as before. The only difference in this setup is the newly added portion of the dataset. **Baseline:** Lastly, for each setup, we train the utility and privacy classifiers on the real (not synthetic) version of the $G$ as well as $V + G$ setup, as the task baseline.

**Experiment Details:** Despite the common practice to use only the front face images in studies, in our work we report the performance on both front and profile faces. In the `Supplementary Material`, you can find the performance of some of our experiments using only front faces. We use 1xH100 GPU to train models and generate synthetic images. For each method, training the LoRA layers took 2 Hours (23 GB), image generation 4 Hours (13 GB), and the classifiers 0.5 Hour. We appended the extensive details of our experiment setups, hyperparameters, and computational complexity of our models in the `Supplementary Material`.

## 4.3 Results and Discussion

We perform our experiments as detailed out in Sec. 4.2, and report the classification accuracy in Tab. 2 to 7. Similar as before, in this section, when we mention $G$ it means the synthetic decoupled version of this split; otherwise we clarify it by noting it as $G_{real}$. The $V + G$ refers to experiments where we use $V$ in addition to $G$ for synthetic image generation (see "Other Experiments" under Sec. 4.2 for more details).

**Impact of the amount of data:** In decoupling methods, the effect of amount of data is twofold; first, it affects the training/decoupling process; and second, it effects the privacy-utility metrics while training the privacy-utility classifiers. In Tab. 1 and. 4 we observe in setup $G$, the small amount of data for training the privacy-utility classifiers, results in overfitting. However, once the classifiers have more training data available $V + G$, the baseline performance on real data is increased, and the privacy leakage drops (20% is random and ideal performance with five race classes) demonstrating larger gap with baseline. This emphasizes on the importance of having access to enough amount of data and clearly defining the splits while reporting privacy-utility performance of decoupling methods. We additionally experimented with a setup where we increased the amount of synthetic

Table 4: **Setup A**: Trained UTKFace and Tested on UTKFace dataset. Classification accuracy percentage for privacy leakage and utility.

| | Method | Privacy (PT, Race, ↓) | Privacy (CLS,Race,↓) | Util (CLS,Age,↑) | Util (CLS,Gender, ↑) |
|---|---|---|---|---|---|
| G | Baseline (Real) | − | 72.30 | 49.47 | 86.19 |
| | CE | 13.69 | 62.63 | 41.17 | 81.72 |
| | CE + Mask | 7.67 | 63.64 | 40.21 | 83.55 |
| | Metric Learning | − | 63.81 | 43.50 | 82.82 |
| V+G | Baseline (Real) | − | 75.79 | 51.1 | 86.7 |
| | CE | 13.82 | 64 | 39.99 | 82.90 |
| | CE + Mask | 7.64 | 64.85 | 41.45 | 83.91 |
| | Metric Learning | − | 59.73 | 41.56 | 83.69 |

Table 5: **Setup B**: Trained UTKFace and Tested on FairFace dataset. Classification accuracy percentage for privacy leakage and utility.

| | Method | Privacy (PT, Race, ↓) | Privacy (CLS,Race,↓) | Util (CLS,Age,↑) | Util (CLS,Gender, ↑) |
|---|---|---|---|---|---|
| G | Baseline (Real) | − | 50.12 | 36.60 | 72.94 |
| | CE | 26.03 | 42.21 | 29.95 | 72.99 |
| | CE + Mask | 26.25 | 44.14 | 33.29 | 73.34 |
| | Metric Learning | − | 43.15 | 34.28 | 70.60 |
| V+G | Baseline (Real) | − | 55.86 | 39.45 | 74.69 |
| | CE | 26.09 | 34.5 | 29.92 | 73 |
| | CE + Mask | 26.12 | 38.14 | 29.48 | 71.61 |
| | Metric Learning | − | 26.32 | 32.09 | 74.35 |

data by generating images from $Tr$ split (i.e., $Tr + V + G$). These experiments confirm our finding regarding the amount of data (see `Supplementary Material`).

**Privacy (PT) vs Privacy (C):** Although, one can use only PT to report the privacy performance, our tables demonstrate, only the PT measure will not be sufficient. In most of the setups, PT reports a perfect privacy number ($\sim 20\%$); however, once the privacy classifier is trained on the generated dataset, the performance is not as perfect. We believe this is due to information leakage during the generation process. It is important to note that while training the LoRA adapters, they make for only $\sim 4.5 - 4.7\%$ of the total number of parameters. However, when these adapters are loaded during the sampling and generation process, the rest of the pretrained (non- LoRA layers) contribute to reconstructing the original signal from the initial image and text prompts. This information leakage hinders the privacy preservation of race (sensitive attribute). This issue can be mitigated by training a full model from scratch which raise computation limitations, specially for training LDMs.

**Cross Generalization:** Tab. 3 and. 5 show the result of our cross-generalization when the model is trained on one dataset (e.g., FairFace dataset) and tested on another dataset (e.g., UTKFace). As we observe (Tab. 3 VS. 2 and Tab. 5 VS. 4), across both $G$ and $V + G$ setups, the privacy and utility performances are comparable, meaning the methods can generalize well to unseen data. We observe more degradation in utility performance, which is expected, given the distribution differences between the two datasets. In. 6 and. 7 we attempt at improving the cross generalization by training/decoupling the base model using data from both FairFace and UKTFace datasets. The joint training, expectedly, results in improvement in baseline performance and CE, and CE + Mask methods when tested on FairFace and UTKFace datasets, specially in $V + G$ setups.

**Privacy-Utility Trade off:** We report the utility performance for non-sensitive attributes using the classification accuracy of utility classifier. Compared to the baseline, in all setups, the utility is persevered. Note that the base model training is based on denoising and preserving the privacy. Therefore, a slight drop in utility is expected. Additionally, it is important to note that the Age attributes had 9 classes, therefore it is a more difficult task compared to gender classification; thus the lower baseline and utility performance.

Table 6: **Setup C**: Trained (Fairface + UTKFace) and Tested on Fairface dataset. Classification accuracy percentage for privacy leakage and utility.

| | Method | Privacy (PT, Race, ↓) | Privacy (CLS,Race,↓) | Util (CLS,Age,↑) | Util (CLS,Gender, ↑) |
|---|---|---|---|---|---|
| G | Baseline (Real) | – | 55.05 | 40.46 | 78.29 |
| | CE | 26.03 | 26.48 | 35.12 | 75.87 |
| | CE + Mask | 14.18 | 31.1 | 35.61 | 74.82 |
| | Metric Learning | – | 38.82 | 34.44 | 72.5 |
| V+G | Baseline (Real) | – | 58.21 | 41.95 | 76.88 |
| | CE | 26.41 | 36.68 | 33.5 | 75.14 |
| | CE + Mask | 14.59 | 45.94 | 34.44 | 76.02 |
| | Metric Learning | – | 50.07 | 37.7 | 72.11 |

Table 7: **Setup C**: Trained (Fairface + UTKFace) and Tested on UTKFace dataset. Classification accuracy percentage for privacy leakage and utility.

| | Method | Privacy (PT, Race, ↓) | Privacy (CLS,Race,↓) | Util (CLS,Age,↑) | Util (CLS,Gender, ↑) |
|---|---|---|---|---|---|
| G | Baseline (Real) | – | 75.82 | 49.89 | 87.12 |
| | CE | 13.69 | 52.14 | 40.38 | 85.77 |
| | CE + Mask | 16.87 | 55.01 | 40.72 | 84.65 |
| | Metric Learning | – | 53.01 | 42.94 | 84.48 |
| V+G | Baseline (Real) | – | 77.67 | 51.04 | 87.94 |
| | CE | 13.88 | 51.07 | 36.02 | 83.21 |
| | CE + Mask | 16.70 | 50.67 | 40.64 | 86.47 |
| | Metric Learning | – | 64.29 | 40.78 | 84.45 |

**Differential Privacy Experiment:** It is important to note that Deco-Bench is flexible and it is readily possible to add Differential Privacy (DP) module on top of any of the methods and offer a formal privacy guarantee. We include the results in the `Supplementary Material`.

**Synthetic and Real Data Distribution:** We measure the quality of generated images using FID score computed between synthetic and real images from $G_{real}$ and $G$ for CE and CE + Mask protocols. The FID scores in Tab. 8 shows a discrepancy between the distribution of real and generated data. We pose this is due to base model (and in turn generation) model is pretrained on laion-aesthetics dataset [30] and then $\sim 4.5\%$ of LoRA parameters are not enough to shift the generation distribution. Fig. 3 shows some examples of our generated images.

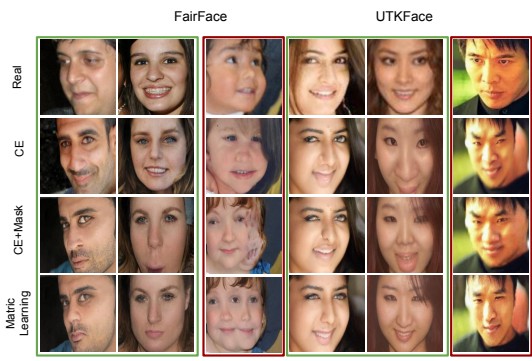

Figure 3: Generated Examples. See `Supplementary` for better resolution.

## 5 Related work

In this section, we discuss prior work on synthetic data and fairness; benchmark datasets and type of data release techniques for protecting sensitive information.

**Synthetic Data and Fairness.** Synthetic data, created using techniques like generative adversarial networks (GANs) [16], variational autoencoders (VAEs) [28, 48], Stable Diffusion (SD) [22, 42], procedural generation methods [7], or simulations [8], emulates real-world data while avoiding privacy concerns. This data is crucial for training models when real data is scarce or sensitive [13]. However, the domain gap and potential biases in synthetic data generation pose challenges [6]. Fairness techniques strive to make models unbiased towards protected groups, often through censoring

Table 8: FID Computation on synthetic vs real G split (Lower score is better).

| Model | Dataset (G set) | Method | UTKFace (Real) | FairFace (Real) |
|---|---|---|---|---|
| FairFace | FairFace | CE | – | 37.24 |
| | UTKFace | CE | 32.06 | – |
| | FairFace | CE + Mask | – | 19.11 |
| | UTKFace | CE + Mask | 17.77 | – |

sensitive information [10, 54, 51, 5, 55, 11]. Our benchmark focuses on anonymizing sensitive data rather than censoring it, ensuring that models maintain original data characteristics while protecting privacy.

**Benchmark Datasets.** Over the past two decades, several real-world datasets have been made publicly available for evaluating the utility of privacy-preserving ML algorithms. Key datasets include the US Census Bureau [3], Colorado (NIST, Differential Privacy Synthetic Data Challenge, 2018) dataset [41, 9], Medical Information Mart for Intensive Care III (MIMIC-III) [26], n2c2: National NLP Clinical Challenges [19], and mobility [2]. While these datasets have been used for various privacy-preserving techniques, our goal is to build a benchmark specifically for decoupling techniques in computer vision tasks. Unlike previous benchmarks focused on synthetic data generation [50] or federated learning [24], we emphasize benchmarking privacy on anonymizing sensitive data.

**Types of Synthetic Data Release.** The release of privacy-preserving data can be broadly categorized into two types: task-specific and task-agnostic. Task-specific techniques transform data to suit a specific task, ensuring the data is optimized for that particular purpose. In contrast, task-agnostic techniques share data in a non-interactive manner, making the data suitable for a variety of tasks without prior knowledge of the specific application. Task-specific data release techniques [14, 15, 4] often use central differential privacy (DP) to transform data for specific tasks, like answering aggregate queries or training models. Recent advancements in adversarial learning have led to task-specific latent representations to protect sensitive information [32, 52, 17, 44, 31, 37, 38, 45, 35, 47]. Our work focuses on task-agnostic techniques that share data non-interactively, similar to local DP [40]. By integrating various decoupling techniques [48, 11, 47], our unified system benchmarks these methods, evaluating their privacy-utility trade-offs. Our framework releases source code, pre-trained models, and datasets of decoupled representations to foster further research in privacy-preserving ML.

## 6  Conclusion

In this work, we took a step toward benchmarking decoupling techniques and evaluating their performance within a systematic framework. We emphasized the importance of clearly defining every aspect of evaluation. We tested our framework and demonstrated a step-by-step protocol for training and evaluating decoupling techniques. We want to emphasize, although we use face images with certain attributes as our use-case; the same principal and methods stand for other types of data and attributes. **Future Work:** In our current study, we distributed and added the LoRA layers in a balanced way across the backbone. However, one can study and find the most effective layers for adding the LoRA weights considering privacy-utility trade off. Since most of the human attributes include rather high frequency features this task is not trivial and requires in depth study of internal diffusion mechanisms. Additionally, it is desirable to add stronger anonymization methods taking into consideration regulations such as the General Data Protection Regulation (GDPR) and add interpretability for privacy leakage. Lastly, it is of interest to add attack/defense mechanisms to our framework and study the interaction of each attack mechanism with the decoupling algorithm.

**Broader Impact:** We hope this research encourages future efforts to enhance privacy protection in digital media and guides the ethical use and development of privacy preserving image generation technologies. This benchmark is designed for the empirical evaluation of decoupling algorithms. Consequently, any conclusions drawn from the benchmark results should also consider the theoretical and worst-case guarantees of the algorithms. Since our dataset is compiled exclusively from publicly accessible, non-private sources, we do not anticipate any privacy issues related to its release.

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

# A   Appendix / supplemental material

In this section, we present our supplementary materials that expand on details of our experiments and further discuss our results. In Sec. A.1 we describe our $Tr + V + G$ setup and present the results. Sec. A.2 focuses on our experiments using only front faces of each dataset. In Sec. A.3, we elaborate on our setups and hyperparameters in details. And finally, Fig. 4 presents the high-resolution equivalent of Fig. 3 from the main paper.

## A.1   Tr+V+G Experiments

In the main section of the paper, we observed the beneficial effect of adding $V$ split while training the privacy and utility classifiers (i.e., $V + G$ vs $G$). As we discussed in the main paper, the increased amount of data during classifiers training helps alleviate overfitting and allows for a better observation of the privacy-utility trade-off. To further investigate this, we generated synthetic images from the $Tr$ split and trained the classifiers. Tab. 9 to Tab. 14 extend the results from Tab. 2 to Tab. 7 respectively. We observe that only some setups demonstrate improvement. We suggest this may be due to the fact that, despite the significant increase in data, we used the same backbone as before, ResNet18, which might not have sufficient capacity to benefit from the additional $Tr$ set. It is also important to note, that since the base model is trained on $Tr$, we do not evaluate the privacy using PT for this setup.

Table 9: **Setup A**: Trained Fairface and Tested on Fairface dataset. Classification accuracy percentage for privacy leakage and utility.

| | Method | Privacy (PT, Race, ↓) | Privacy (CLS,Race,↓) | Util (CLS,Age,↑) | Util (CLS,Gender, ↑) |
|---|---|---|---|---|---|
| Tr+V+G | Baseline (Real) | – | 62.38 | 44.8 | 83.36 |
| | CE | – | 51.03 | 32.72 | 80.25 |
| | CE + Mask | – | 48.87 | 38.09 | 79.81 |
| | Metric Learning | – | 54.89 | 29.56 | 76.47 |

Table 10: **Setup B**: Trained FairFace and Tested on UTKFace dataset. Classification accuracy percentage for privacy leakage and utility.

| | Method | Privacy (PT, Race, ↓) | Privacy (CLS,Race,↓) | Util (CLS,Age,↑) | Util (CLS,Gender, ↑) |
|---|---|---|---|---|---|
| Tr+V+G | Baseline (Real) | – | 81.24 | 56.02 | 90.55 |
| | CE | – | 69.71 | 46.65 | 85.97 |
| | CE + Mask | – | 61.22 | 44.35 | 87.2 |
| | Metric Learning | – | 61.33 | 39.31 | 84.79 |

Table 11: **Setup A**: Trained UTKFace and Tested on UTKFace dataset. Classification accuracy percentage for privacy leakage and utility.

| | Method | Privacy (PT, Race, ↓) | Privacy (CLS,Race,↓) | Util (CLS,Age,↑) | Util (CLS,Gender, ↑) |
|---|---|---|---|---|---|
| Tr+V+G | Baseline (Real) | – | 81.24 | 56.02 | 90.55 |
| | CE | – | 61.36 | 41.23 | 86.16 |
| | CE + Mask | – | 73.31 | 38.95 | 87.82 |
| | Metric Learning | – | 61.75 | 45.16 | 86.81 |

## A.2   Front Face Experiments

In many studies utilizing datasets of face images, it is common practice to use only front faces. In our main paper, we presented results using both front and profile faces. To accommodate a variety of experimental setups in our benchmark and to study the effect of using only front faces on the privacy-utility trade-off, we repeat our experiments in this section using only front faces.

Table 12: **Setup B**: Trained UTKFace and Tested on FairFace dataset. Classification accuracy percentage for privacy leakage and utility.

| | Method | Privacy (PT, Race, ↓) | Privacy (CLS,Race,↓) | Util (CLS,Age,↑) | Util (CLS,Gender, ↑) |
|---|---|---|---|---|---|
| Tr+V+G | Baseline (Real) | – | 62.38 | 44.8 | 83.36 |
| | CE | – | 49.7 | 36.03 | 77.38 |
| | CE + Mask | – | 46.96 | 36.29 | 78.87 |
| | Metric Learning | – | 52.05 | 36.34 | 78.97 |

Table 13: **Setup C**: Trained (Fairface + UTKFace) and Tested on FairFace dataset. Classification accuracy percentage for privacy leakage and utility.

| | Method | Privacy (PT, Race, ↓) | Privacy (CLS,Race,↓) | Util (CLS,Age,↑) | Util (CLS,Gender, ↑) |
|---|---|---|---|---|---|
| Tr+V+G | Baseline (Real) | – | 65.74 | 44.53 | 84.63 |
| | CE | – | 46.91 | 39.42 | 81.45 |
| | CE + Mask | – | 56.35 | 38.46 | 79.89 |
| | Metric Learning | – | 57.92 | 41.27 | 80.9 |

Table 14: **Setup C**: Trained (Fairface + UTKFace) and Tested on UTKFace dataset. Classification accuracy percentage for privacy leakage and utility.

| | Method | Privacy (PT, Race, ↓) | Privacy (CLS,Race,↓) | Util (CLS,Age,↑) | Util (CLS,Gender, ↑) |
|---|---|---|---|---|---|
| Tr+V+G | Baseline (Real) | – | 81.81 | 54.84 | 91 |
| | CE | – | 62.01 | 40.75 | 88.19 |
| | CE + Mask | – | 57.73 | 42.35 | 88.92 |
| | Metric Learning | – | 66.28 | 49.49 | 87.46 |

#### A.2.1 Front Face Detection:

To detect images with front faces, we processed all splits of the FairFace and UTKFace datasets using dlib library [27]. We perform front face detection both in original image size and also in $128 \times 128$ resolution; given this is the image resolution we train our base model with. Below, we present the statistics of front faces in each data split of the FairFace and UTKFace datasets. As observed, the FairFace dataset contains more profile images compared to the UTKFace dataset.

Table 15: Dataset Statistics for Front Faces of FairFace and UTKFace datasets. The number of classes for the datasets after cross label mapping are demonstrated (see Sec.4.1, main paper).

| Dataset | Train (Tr) Samples | Val (V) Samples | Gen (G) Samples | Test Samples | Privacy Attr. (#Classes) | Utility Attr. (#Classes) | Eval Metric |
|---|---|---|---|---|---|---|---|
| FairFace | 11263 | 1256 | 4480 | 2747 | Race (C=5) | Age (C=9), Gender (C=2) | Accuracy |
| UTKFace | 12374 | 1372 | 5307 | 3478 | Race (C=5) | Age (C=9), Gender (C=2) | Accuracy |

#### A.2.2 Front Face Evaluation:

In our first round of experiments, we maintain the entire pipeline from the main paper, with the exception of using only front faces in the $Test$ split. Specifically, we train the base model on $Tr$ using both front and profile images. We then generate synthetic images from $G_{real}$, and $V, Tr$ splits using both profile and front images and train the privacy and utility classifiers on them. In the final step, we use only front faces in $Test$ split for evaluation and report the privacy leakage and utility performance. Similarly, for PT evaluation, we use only front faces. The results are presented in Tab. 16 to Tab. 19.

Table 16: **Setup A** : Trained Fairface and Tested on Fairface dataset Front Faces only. Classification accuracy percentage for privacy leakage and utility.

| | Method | Privacy (PT, Race, ↓) | Privacy (CLS,Race,↓) | Util (CLS,Age,↑) | Util (CLS,Gender, ↑) |
|---|---|---|---|---|---|
| **G** | Baseline (Real) | − | 57.66 | 39.86 | 76.01 |
| | CE | 18.71 | 42.34 | 35.86 | 78.59 |
| | CE + Mask | 15.13 | 47.11 | 37.31 | 72.12 |
| | Metric Learning | − | 50.86 | 32.65 | 73.79 |
| **V+G** | Baseline (Real) | − | 58.54 | 41.9 | 78.05 |
| | CE | 18.56 | 49.91 | 36.55 | 75.83 |
| | CE + Mask | 15.34 | 50.24 | 37.6 | 80.42 |
| | Metric Learning | − | 53.91 | 37.57 | 76.37 |
| **Tr+V+G** | Baseline (Real) | − | 67.24 | 49.33 | 87.08 |
| | CE | − | 45.94 | 40.84 | 82.34 |
| | CE + Mask | − | 28.76 | 36.59 | 82.64 |
| | Metric Learning | − | 53.66 | 40.01 | 82.71 |

Table 17: **Setup B** : Trained Fairface and Tested on UTKFace dataset Front Faces only. Classification accuracy percentage for privacy leakage and utility.

| | Method | Privacy (PT, Race, ↓) | Privacy (CLS,Race,↓) | Util (CLS,Age,↑) | Util (CLS,Gender, ↑) |
|---|---|---|---|---|---|
| **G** | Baseline (Real) | − | 73.23 | 48.33 | 86.83 |
| | CE | 42.87 | 66.1 | 42.9 | 81.94 |
| | CE + Mask | 16.90 | 52.04 | 44.31 | 84.39 |
| | Metric Learning | − | 59.8 | 39.07 | 83.58 |
| **V+G** | Baseline (Real) | − | 72.94 | 50.72 | 85.48 |
| | CE | 42.82 | 69.58 | 44.77 | 84.91 |
| | CE + Mask | 16.74 | 57.82 | 39.59 | 84.68 |
| | Metric Learning | − | 61.18 | 43.16 | 83.84 |
| **Tr+V+G** | Baseline (Real) | − | 81.68 | 56.41 | 91.14 |
| | CE | − | 69.47 | 46.75 | 86.06 |
| | CE + Mask | − | 61.33 | 44.51 | 87.9 |
| | Metric Learning | − | 62.13 | 39.25 | 84.85 |

Table 18: **Setup C**: Trained Fairface+UTKFace Tested on FairFace dataset Front Faces only. Classification accuracy percentage for privacy leakage and utility.

| | Method | Privacy (PT, Race, ↓) | Privacy (CLS,Race,↓) | Util (CLS,Age,↑) | Util (CLS,Gender, ↑) |
|---|---|---|---|---|---|
| **G** | Baseline (Real) | − | 56.28 | 43.54 | 81.43 |
| | CE | 26.25 | 48.02 | 36.33 | 79.87 |
| | CE + Mask | 15.13 | 46.81 | 35.57 | 76.3 |
| | Metric Learning | − | 45.61 | 28.69 | 73.97 |
| **V+G** | Baseline (Real) | − | 59.23 | 47.36 | 83.15 |
| | CE | 26.87 | 50.16 | 37.02 | 80.31 |
| | CE + Mask | 15.34 | 42.81 | 38.26 | 79.32 |
| | Metric Learning | − | 56.13 | 33.71 | 78.01 |
| **Tr+V+G** | Baseline (Real) | − | 71.06 | 50.38 | 87.19 |
| | CE | − | 57.41 | 42.88 | 82.71 |
| | CE + Mask | − | 55.08 | 41.35 | 80.82 |
| | Metric Learning | − | 59.56 | 43.17 | 83.4 |

Table 19: **Setup C**: Trained Fairface+UTKFace Tested on UTKFace dataset Front Faces only. Classification accuracy percentage for privacy leakage and utility.

|  | Method | Privacy (PT, Race, ↓) | Privacy (CLS,Race,↓) | Util (CLS,Age,↑) | Util (CLS,Gender, ↑) |
|---|---|---|---|---|---|
| G | Baseline (Real) | − | 76.57 | 50.86 | 88.18 |
| | CE | 13.64 | 59.49 | 43.42 | 85.51 |
| | CE + Mask | 16.90 | 58.77 | 42.32 | 87.21 |
| | Metric Learning | − | 51.04 | 41.17 | 82.78 |
| V+G | Baseline (Real) | − | 76.88 | 52.67 | 88.04 |
| | CE | 13.82 | 62.1 | 42.21 | 86.83 |
| | CE + Mask | 16.74 | 64.17 | 43.67 | 87.98 |
| | Metric Learning | − | 55.15 | 42.73 | 84.88 |
| Tr+V+G | Baseline (Real) | − | 80.71 | 55.78 | 90.8 |
| | CE | − | 62.39 | 41.09 | 88.13 |
| | CE + Mask | − | 58.05 | 42.73 | 89.13 |
| | Metric Learning | − | 66.53 | 49.74 | 87.61 |

### A.2.3 Front Face Training and Evaluation

In this section, we extend our front face experiments by repeating the entire pipeline using only front faces. Specifically, we train the base model using only front faces of $Tr$ set, followed by generating synthetic images from only front faces of $G_{real}$, $V$ and $Tr$ splits. We then train the privacy and utility classifiers using these generated images and, finally, evaluate the models on only the front faces of $Test$ split. Tab. 20 to Tab. 23 demonstrate the results.

Table 20: **Setup A (Front Faces only)**: Trained Fairface Front Faces only and Tested on Fairface dataset Front Faces only. Classification accuracy percentage for privacy leakage and utility.

|  | Method | Privacy (PT, Race, ↓) | Privacy (CLS,Race,↓) | Util (CLS,Age,↑) | Util (CLS,Gender, ↑) |
|---|---|---|---|---|---|
| G | Baseline (Real) | − | 54.42 | 34.55 | 71.17 |
| | CE | 18.70 | 42.52 | 34.44 | 72.52 |
| | CE + Mask | 15.13 | 46.01 | 35.02 | 72.26 |
| | Metric Learning | − | 51.18 | 36.08 | 72.84 |
| V+G | Baseline (Real) | − | 56.75 | 38.7 | 79.43 |
| | CE | 18.54 | 33.49 | 35.24 | 72.04 |
| | CE + Mask | 15.34 | 40.88 | 37.5 | 76.77 |
| | Metric Learning | − | 45.18 | 36.4 | 76.88 |
| Tr+V+G | Baseline (Real) | − | 67.53 | 45.9 | 85.84 |
| | CE | − | 56.57 | 41.32 | 81.58 |
| | CE + Mask | − | 45.69 | 35.27 | 82.82 |
| | Metric Learning | − | 46.09 | 40.52 | 80.85 |

Table 21: **Setup B (Front Faces only)**: Trained Fairface Front Faces only and Tested on UTKFace dataset Front Faces only. Classification accuracy percentage for privacy leakage and utility.

| | Method | Privacy (PT, Race, ↓) | Privacy (CLS,Race,↓) | Util (CLS,Age,↑) | Util (CLS,Gender, ↑) |
|---|---|---|---|---|---|
| **G** | Baseline (Real) | – | 72.83 | 48.88 | 83.32 |
| | CE | 42.87 | 63.02 | 43.93 | 81.08 |
| | CE + Mask | 16.90 | 52.59 | 38.47 | 86.26 |
| | Metric Learning | – | 60.7 | 34.76 | 83.29 |
| **V+G** | Baseline (Real) | – | 74.24 | 48.91 | 87.72 |
| | CE | 42.82 | 66.07 | 41.35 | 83.84 |
| | CE + Mask | 16.74 | 61.96 | 44.13 | 85.48 |
| | Metric Learning | – | 62.33 | 42.73 | 83.73 |
| **Tr+V+G** | Baseline (Real) | – | 79.61 | 54.17 | 88.9 |
| | CE | – | 61.87 | 46.18 | 84.73 |
| | CE + Mask | – | 65.96 | 47.18 | 87.26 |
| | Metric Learning | – | 63.89 | 44.48 | 89.1 |

Table 22: **Setup C (Front Faces only)**: Trained Fairface + UTKFace Front Faces only and Tested on Fairface dataset Front Faces only. Classification accuracy percentage for privacy leakage and utility.

| | Method | Privacy (PT, Race, ↓) | Privacy (CLS,Race,↓) | Util (CLS,Age,↑) | Util (CLS,Gender, ↑) |
|---|---|---|---|---|---|
| **G** | Baseline (Real) | – | 59.96 | 42.77 | 78.96 |
| | CE | 26.25 | 45.58 | 35.35 | 78.41 |
| | CE + Mask | 15.13 | 51.62 | 38.11 | 77.18 |
| | Metric Learning | – | 42.34 | 31.67 | 77.25 |
| **V+G** | Baseline (Real) | – | 59.05 | 44.05 | 83.29 |
| | CE | 26.87 | 43.79 | 39.79 | 77.9 |
| | CE + Mask | 15.34 | 44.99 | 39.06 | 75.06 |
| | Metric Learning | – | 45.03 | 37.68 | 76.45 |
| **Tr+V+G** | Baseline (Real) | – | 67.09 | 49.22 | 86.71 |
| | CE | – | 50.16 | 38.48 | 80.74 |
| | CE + Mask | – | 51.58 | 41.35 | 82.64 |
| | Metric Learning | – | 57.04 | 43.17 | 81.87 |

## A.3 Experiment Details:

In this section, we provide detailed training information and hyperparameters for every component in our pipeline, supplementing Sec.4.2 of the main paper. **Base model:** to train the base model, we resize all images to $128 \times 128$. The LoRA adapter layers are initialized with Gaussian distribution, and their rank is set to 4. The base model is trained for 15 epochs using a batch size of 128 and the AdamW [33] optimizer with weight decay of $0.01$. The learning rate is $1e-5$ for CE and CE+Mask, and $1e-4$ for Metric learning. We stop training before overfitting occurs. We employ a constant learning rate scheduler with 200 warm-up steps and apply gradient clipping with maximum grad norm of 1. The margin for triplet margin loss (metric learning) is 0.3. The masking ratio for CE+Mask setup is 0.6 and the sigmoid temperature in FGN is 1/30. A LoRA adapter is added to the learnable layer of FGN. The configurations for variational autoencoder and CLIP remain unchanged after loading the pretrained models [1].

**Generation:** During generation, we load our pretrained base model with trained LoRA adaptors into the SD image-to-image pipeline [34]. The strength and guidance scale are set to 0.75 and 7.5, respectively. We fuse the LoRA weights with the scale of 1, meaning the LoRA weights completely replace the weights of the base layer they were added to.

**Classification:** The classifiers are trained using an image resolution of $128 \times 128$ and a batch size of 128. The ResNet18 weights are initialized randomly. We use an SGD optimizer with weight decay of $5e-4$, momentum of $0.9$, and learning rate of $0.1$. An exponential learning rate scheduler with

Table 23: **Setup C (Front Faces only)**: Trained Fairface + UTKFace Front Faces only and Tested on UTKFace dataset Front Faces only. Classification accuracy percentage for privacy leakage and utility.

| | Method | Privacy (PT, Race, ↓) | Privacy (CLS,Race,↓) | Util (CLS,Age,↑) | Util (CLS,Gender, ↑) |
|---|---|---|---|---|---|
| G | Baseline (Real) | – | 76.77 | 52.19 | 86.69 |
| | CE | 13.64 | 63.77 | 39.59 | 85.85 |
| | CE + Mask | 16.90 | 58.8 | 44.88 | 85.34 |
| | Metric Learning | – | 58.05 | 43.3 | 84.73 |
| V+G | Baseline (Real) | – | 73.75 | 51.44 | 88.5 |
| | CE | 13.82 | 67.94 | 44.45 | 85.83 |
| | CE + Mask | 16.74 | 53.54 | 40.91 | 85.34 |
| | Metric Learning | – | 66.5 | 44.42 | 84.96 |
| Tr+V+G | Baseline (Real) | – | 82.58 | 56.12 | 91.17 |
| | CE | – | 64.09 | 45.95 | 87.92 |
| | CE + Mask | – | 64.26 | 48.39 | 87.67 |
| | Metric Learning | – | 67.4 | 44.22 | 88.59 |

gamma of 0.9 is employed, and the models are trained for 100 epochs, stopping training after 20 epochs of no improvement (delta 0.01) in validation loss.

### A.3.1 Computational Complexity:

As mentioned earlier we use 1xH100 GPU. **Training LoRA:** the conditional Unet number of parameters including added LoRA layers is 859.62M from which 99.840K are for trainable LoRA adapters. For CE an additional FC layer is added which adds 1.5K number of parameters (using 128x128 image resolution). In the CE+mask model the filter generation network adds 90K trainable parameters. The overall model including image encoder (VAE) and text encoders with batch size of 128 takes 23 GB GPU memory and 2 Hours to train for 15 epochs. In this setup, the number of FLOPs are: 4.33 TFLOPs for VAE, 2.89 TFLOPs for Unet backbone, and 0.85 TFLOPs for text encoder. **Image Generation:** the image generation pipeline occupies 13 GB of GPU memory and takes 4 Hours to generate images from G split ( 6K images) with batch size of 1, which approximately translates to 1-2 seconds per image on H100 with 74.68 TFLOPs. **Classifiers:** we use ResNet18 (from scratch) to train privacy and utility classifiers which has 11M parameters, 151 GFLOPs and occupy 2.5 GB of GPU memory with batch size of 128. The training on G split takes 0.5 Hours and the inference time is less than 1 minute on the test set.

### A.4 Differential Privacy:

In order to showcase, we used Opacus privacy engine [53] with $epsilon - tolerance = 0.1$, $target - delta = 1e - 5$, and $epsilon = 5$. We add DP on top of CE methods for setup A of FairFace datasets trained on $V + G$ splits. We report the following results:

Table 24: **Setup A**: Trained Fairface and Tested on Fairface dataset. Classification accuracy percentage for privacy leakage and utility.

| | Method | Privacy (PT, Race, ↓) | Privacy (CLS,Race,↓) | Util (CLS,Age,↑) | Util (CLS,Gender, ↑) |
|---|---|---|---|---|---|
| V+G | Baseline (Real) | – | 55.86 | 39.45 | 74.69 |
| | CE | 19.22 | 46.23 | 34.44 | 73.13 |
| | CE + DP | 26.18 | 39.21 | 35.04 | 73.78 |

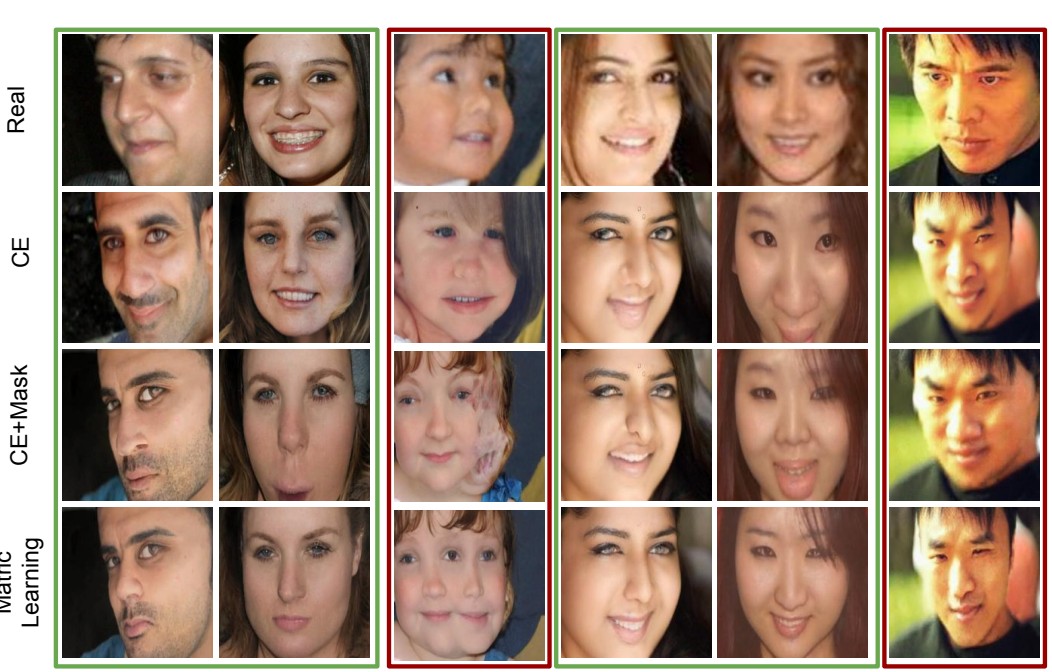

Figure 4: Generated Examples. High resolution image from main paper.

