# OpenReview forum: "DECO-Bench: Unified Benchmark for Decoupled Task-Agnostic Synthetic Data Release"
_NeurIPS.cc/2024/Datasets_and_Benchmarks_Track — NeurIPS 2024 Track Datasets and Benchmarks Poster_

### Official Review · Reviewer_fx2c · 2024-07-12
**Review for DECO-Bench**

**Rating:** 7
**Confidence:** 4
**Correctness:** Yes
**Clarity:** Yes

**Review:**

While the paper presents a well-structured and comprehensive framework for benchmarking decoupled synthetic data methods, it needs to address the limitations in cross-dataset generalization and potential biases introduced by pre-trained models. Additionally, including a discussion on formal privacy guarantees would strengthen the paper's contribution to the field.

**Strengths:**

1. Task and Data Agnostic Design: DECO-Bench's design is versatile, and applicable across various domains and data types. This flexibility ensures that researchers can use the framework for different tasks and datasets without significant modifications, enhancing its utility and relevance in a wide range of scenarios.
2. Comprehensive Evaluation Perspective: The framework evaluates both privacy and data utility, offering a holistic view of the trade-offs involved. This dual focus ensures that data remains useful for practical applications while maintaining privacy, providing a balanced assessment of decoupling techniques.
3. Thorough Experiment Design: DECO-Bench's experimental setup is meticulously crafted, using multiple datasets and detailed evaluation protocols. This thorough approach allows for a deep understanding of the performance, robustness, and generalizability of different decoupling techniques under various conditions.

**Additional Feedback:**

NA

**Documentation:**

Yes

**Limitations:**

Yes

**Opportunities For Improvement:**

1. Lack of Formal Privacy Guarantees: DECO-Bench relies on empirical evaluations without offering formal privacy guarantees like differential privacy methods. It would strength this paper if formal privacy guarantees can be added or clarified.
2. Cross-Generalization Performance: The framework shows significant performance degradation when applied across different datasets (FairFace and UTKFace). This suggests potential limitations in generalizability, as the decoupling techniques may not perform well on unseen datasets with different characteristics.
3. Dependency on Pre-trained Models: DECO-Bench's reliance on pre-trained models, such as LDMs, can cause information leakage and suboptimal privacy protection. These models might not align perfectly with the target dataset, retaining sensitive information and reducing the effectiveness of decoupling.

**Relation To Prior Work:**

Yes

**Summary And Contributions:**

The paper proposes DECO-Bench, a unified benchmark framework for evaluating decoupled task-agnostic synthetic data release methods in privacy-preserving machine learning. The framework integrates various decoupling techniques with synthetic data generation and evaluation protocols including Cross-Entropy Decoupling (CE), Cross-Entropy with Masking (CE + Mask), and Metric Learning. The evaluation session uses Latent Diffusion Models (LDMs) to generate decoupled datasets and benchmarks various decoupling techniques, assessing privacy-utility trade-offs.

---

> ### Author Rebuttal · Authors · 2024-08-16
>
> We really appreciate our reviewer's acknowledgement of our method versatility, flexibility, meticulous experiment design, and detailed evaluation. We ensure to integrate your insightful feedback into our work.
>
> **Q1:  It would strengthen this paper if formal privacy guarantees can be added or clarified.**
>
> A1: Thank you for your suggestion. Per your feedback, we added differential privacy to our setup A **(Table 2)** for the CE model. Please see our global response for results.
>
> **Q2: This suggests potential limitations in generalizability, as the decoupling techniques may not perform well on unseen datasets with different characteristics.**
>
> A2: It is true that there is a performance degradation visible in cross-generalization setup. However, based on our observations, we believe this is due to significant distribution mismatch between FairFace and UTKFace.
>
> + **Image quality:** In **Table 6** we observe a better FID score (lower) for UTKFace across both CE and CE+mask setup. Additionally, from our front face detection study **(Table 11 of Supplementary material)** for the FairFace training set we detect 4.4K profile face images and for UTKFace we only detect 0.3K profile face images.
> + **UTKFace helps FairFace:** we observe when UTKFace is added to the FairFace dataset during training, it helps the performance on the FairFace dataset. We can see that by comparing V+G in setup A **(Table 2)** and V+G in setup C **(Table 4)**; that by adding UTKFace during training, the privacy and utility performances improve across most methods. This might be due to the fact that UTKFace has “better” or “more variant” faces. Please note, we chose V+G setup so the amount of data does not interfere with cross generalization analysis.
> + Additionally, following your observation we include another set of cross-generalization experiment. We observe the same effect when we cross generalize by training on UTKFace and evaluating on FairFace **(similar setup to Table 3 with datasets switched)**. We observe most of the results are comparable with **Table 2**, meaning by training on UTKFace we are able to decouple FairFace.
>
> ##### Trained on UTKFace tested on FairFace
>
> | Method (G)| Privacy (CLS, Race) | Utility (CLS,Age)| Utility (CLS,Gender) |
> | ------ | ------| ------ |------ |
> | CE | 42.21 |  29.95 | 72.99|
> | CE+Mask   | 44.14   | 33.29  | 73.34|
> | Metric Learning   | 43.15  | 34.28 | 70.60|
>
> | Method (V+G)| Privacy (CLS, Race) | Utility (CLS,Age)| Utility (CLS,Gender) |
> | ------ | ------| ------ |------ |
> | CE | 34.5  |  29.92 | 73|
> | CE+Mask   | 38.14   | 29.48  | 71.61|
> | Metric Learning  | 26.32  | 32.09 | 74.35|
>
> Given the above discussion, we believe the cross-generalization performance degradation is mostly due to great distribution mismatch across these two datasets, especially affecting cross generalization from FairFace to UTKFace. Therefore, as pointed out, it improves the performance if the dataset characteristics are rather similar. Thank you for your feedback. we ensure to include this table and analyses.
>
> **Q3: Reliance on pre-trained models, such as LDMs, can cause information leakage and suboptimal privacy protection.**
>
> A3: As discussed in **line 288-297**, we also observed the information leakage, when comparing the discrepancy between PT evaluation and CLS privacy evaluation. It means, when we run classification inference on _G_ split using the trained model with LoRA weights (PT evaluation), the privacy is almost perfect around 20% across the setups.  However, when we generate synthetic images from G_real split and train our privacy classifier on them and test on the Test split (CLS evaluation), the privacy performance degrades (in comparison to PT). We can observe this effect even across the setups where the amount of data is not a concern; which leaves us with the possibility of information leakage.
>
> We recommend the below workaround to address this concern:
>
> + Differential privacy: one workaround, also per your previous suggestion, is to include DP on top of the decoupling methods to enhance privacy.
> + Training from scratch: our framework is flexible to take any pre-trained model. Therefore, one can train their own SD model from scratch using a dataset they have consent for and the efficient available training techniques. They then can continue using our framework for the rest of the pipeline. We have this in mind as our future work.
> + Finding the most effective layer: In our current study, we distributed and added the LoRA layers in a balanced way across the backbone. However, one can study and find the most effective layers for adding the LoRA weights considering privacy-utility trade off, and add the layers where most information leakage is happening. It is important to note, since most of the human attributes include rather high frequency and detailed information (i.e., ethnicity) this task is not trivial and requires in depth study of internal diffusion mechanism and generation pipeline.
>
> Thank you for your observation. We will add this analysis to our text.

---

> > ### Author Response · Authors · 2024-08-23
> > **Friendly Reminder**
> >
> > Dear Reviewer fx2c,
> >
> > This is a very gentle reminder that we are approaching the midpoint of the discussion period. Your comments and suggestions have been crucial in helping improve the quality of our paper, and we are truly grateful for your contributions.
> > We have tried our best to address your concerns through our response. Specifically, we have:
> >
> > + Added differential privacy to one of our setups and reported/discussed the results.
> > + Elaborated on cross generalization performance by adding UTKFace to FairFace cross generalization experiments and discussing our findings.
> > + Discussed workarounds regarding the usage of pretrained models and information leakage.
> >
> > We do hope to have your feedback and look forward to answering any additional questions you have. Thank you very much.
> >
> > Best regards,
> >
> > The Authors

---

### Official Review · Reviewer_mYev · 2024-07-23
**Reviews from Reviewer mYev**

**Rating:** 7
**Confidence:** 4
**Correctness:** Yes.
**Clarity:** Yes.

**Review:**

### Strengths:

1. The paper introduces a novel and comprehensive benchmarking framework, DECO-Bench, which integrates decoupling techniques with synthetic data generation and evaluation metrics. This framework fills a significant gap in the field by providing a systematic method for evaluating privacy-preserving techniques.
2. The authors conduct extensive experiments across multiple datasets, including FairFace and UTKFace, to demonstrate the effectiveness and generalizability of their framework. The results are well-documented and show clear improvements in privacy-utility trade-offs.
3. The paper's contributions are substantial, including the development of a new benchmark, the release of decoupled datasets, and the integration of various decoupling methods. These contributions are valuable for the research community and can accelerate advancements in privacy-preserving machine learning.
4. The paper is well-structured and clearly presents the methodology, experiments, and results.

### Weaknesses:
1. While the experimental setup is generally well-explained, some sections could benefit from additional clarity. For instance, the details on the implementation of specific decoupling techniques and their integration with the image generation pipeline could be expanded.
2. The two-stage approach and the integration of decoupling techniques with synthetic data generation may introduce additional computational overhead. An analysis of the computational complexity and potential optimization strategies would be beneficial.

### Minor Issues:
1. Line 9: “evaluating” should be “evaluate”
2. Line 20: “Imagine a party owns” should be “Imagine a party owning”

**Strengths:**

The contribution of this paper is significant and highly relevant to the privacy-preserving machine learning community.

**Additional Feedback:**

No.

**Documentation:**

Yes.

**Ethics:**

No ethic issues.

**Limitations:**

The authors have discussed the limitations.

**Opportunities For Improvement:**

The layout and aesthetic quality of the figures and tables require refinement for better presentation.

**Relation To Prior Work:**

Yes.

**Summary And Contributions:**

This paper addresses the critical problem of privacy preservation in machine learning datasets by proposing a comprehensive benchmarking framework for task-agnostic decoupling methods. The authors introduce a unified system that integrates various decoupling techniques with synthetic data generation and evaluation protocols. Their primary contribution is the DECO-Bench framework, which systematically evaluates the privacy-utility trade-offs of different decoupling methods. The paper presents extensive experiments demonstrating the framework's effectiveness using several visual datasets, particularly focusing on image classification tasks. The authors also release their source code, pre-trained models, and datasets to facilitate further research in privacy-preserving machine learning.

---

> ### Author Rebuttal · Authors · 2024-08-16
>
> We really appreciate our reviewer acknowledging the value, novelty, and comprehensiveness of our framework to fill a significant gap for evaluating decoupling methods. We ensure to integrate your insightful feedback into our work.
>
> **Q1: The details on the implementation of specific decoupling techniques and their integration with the image generation pipeline could be expanded.**
>
> A1: As discussed in **line 59-86** and **line 196-212**, for all the three methods, CE, CE+mask, and Metric learning, our first step is to include the LoRA layers in the conditional Unet backbone of the SD model. To do this, we take every cross attention block and add the LoRA adapters to the last conv projection layer of that block (right at the output of the block). We apply these layers in a balanced manner across the down, mid, and upsampling blocks. During training in addition to finetune the model on training split of our dataset in hand, our goal is to restructure the learned latent space, such that the sensitive attribute is anonymized. To achieve this, we average the output of these LoRA layers (LoRA features) to handle varying resolution across cross attention layers of conditional Unet and follow the methods below.
>
> + **CE:** we feed LoRA features to an FC layer for classification of sensitive attributes, followed by a CE loss. During training, we **maximize** this classification CE loss while minimizing the diffusion MSE loss. The intuition behind this method is obfuscating the latent space such that the sensitive attribute cannot be discriminated using the latent representations. **(line 87-92 and line 215-219).**
>
> + **CE+mask:** this method is built upon the CE, with an additional dynamic filter generation network. We add a trainable FC layer right after the first LoRA adapter, that outputs a feature map score. These feature map scores are weakly discretized (using temperature sigmoid) and thresholded to output a binary mask. Finally the binary mask is applied to the output of the first LoRA layer which is passed down to the rest of the model. The ratio of masked channels is a hyperparameter. The idea of CE+mask is to apply a data driven channel pruning method such that the sensitive information is selectively obfuscated. **(line 93-97 and line 219-222).**
>
> + **Metric Learning:** supervised metric learning aims to learn a metric such that the samples with same labels are brought closer together and pushed apart from samples with different labels. However, we seek the opposite effect; meaning, we aim to bring different labels of sensitive attributes (e.g., ethnicity) closer together in the latent space such that the latent space cannot be used to discriminate between them. Therefore, we use a semi-hard negative and positive miner to retrieve the negative pairs and positive pairs. We make a triplet where the anchor remains the same, while we swap negative and positive pairs. These triplets are input to a triple margin loss which is minimized as usual. What this mechanism does is mapping different sensitive attributes close together in the latent space and making the discrimination difficult, therefore preserving privacy of sensitive attributes **(line 98-105 and line 222-227).**
>
> + Once the training is complete using each of these methods, we save the LoRA weights and load them in the generation pipeline. For each layer that we created a LoRA adapter, we completely replace the weights with LoRA weights. We use each image-prompt pair from G_real data split, to generate a new image conditioned on this pair. The generated images are to retain the non-sensitive attributes and anonymize the sensitive one (e.g., race) **(line 228-236).**
>
> **Q2: An analysis of the computational complexity and potential optimization strategies would be beneficial.**
>
> A2: Thank you for your observation. Per your feedback, we added the computational complexity of our models. Please see our global response for details.
>
> + **Optimization strategy:** one strategy is to save on number of new parameters by finding where in the model is the most effective to insert the LoRA adapters. In our current study, we distributed and added the LoRA layers in a balanced way across the backbone. However, one can study and find the most effective layers for adding the LoRA weights considering privacy-utility trade off. Using this method, we could possibly use less LoRA layers and reduce the number of parameters.
>
> + **Efficient inference and reduced memory:** aside from the above strategy, HuggingFace offers memory reducing techniques such as CPU offloading, memory-efficient attention, gradient accumulation; as well as methods to speed up inference such as mixed precision training. The combination of these approaches, in addition to parameter efficient fine tuning, working with lower resolution images, and limiting the number of inference steps will lower the computational complexity.

---

> > ### Author Response · Authors · 2024-08-23
> > **Friendly Reminder**
> >
> > Dear Reviewer mYev,
> >
> > This is a very gentle reminder that we are approaching the midpoint of the discussion period. Your comments and suggestions have been crucial in helping improve the quality of our paper, and we are truly grateful for your contributions.
> > We have tried our best to address your concerns through our response. Specifically, we have:
> >
> > + Elaborated on the details of our work, specifically, the internal mechanism of the decoupling algorithms.
> > + Added analysis of computational complexity and suggested some optimization strategies.
> >
> > We do hope to have your feedback and look forward to answering any additional questions you have. Thank you very much.
> >
> > Best regards,
> >
> > The Authors

---

### Official Review · Reviewer_NgJu · 2024-08-09
**Limited benchmark contribution and evaluation**

**Rating:** 5
**Confidence:** 3
**Correctness:** The claims are generally correct.
**Clarity:** The paper is generally well written.

**Review:**

**Quality.** The paper's quality is somehow below the acceptance bar.  Specifically, the dataset decoupling is not well described and how the benchmark is set up is not clear from the paper (mainly Section 3). Moreover, there is no systematic evaluation of previous dataset decoupling method and the evaluation is only conducted on FairFace and UTKFace, which is limited to conclude the generalization to other tasks.

**Clarify.** The paper is generally well written and easy to follow.


**Significance.** The paper is, as far as I know, the first benchmark study for the dataset decoupling. The dataset coupling is an interesting solution for data privacy for training machine learning models.

**Pros.**
1. The paper is generally well written and easy to follow.
2. The first attempt to build a benchmark for dataset decoupling methods.

**Cons.**
1. The benchmark analysis and contribution is not sufficient.
2. The evaluation is limited: it is only conducted on two facial attribute datasets.

**Strengths:**

* First attempt to benchmark dataset decoupling methods.

**Additional Feedback:**

None

**Documentation:**

The benchmark should contain more details such as the common part of different methods and what are their unique advantages.

**Limitations:**

No, there is no potential negative social impact.

**Opportunities For Improvement:**

* Include more thorough literature review of dataset decoupling method.
* Conduct more evaluation of different types of datasets.
* Clarify the benchmark organizations and brings some constructive takeaway for future work.

**Relation To Prior Work:**

Yes, the paper differs from previous contributions. However, I would suggest the authors to discuss more about the dataset condensation and how it relates to the idea of decoupling when it comes to the privacy. For example, here are some related work [1-5].

[1] Dataset Condensation with Distribution Matching. WACV 2023.

[2] Privacy for Free: How does Dataset Condensation Help Privacy?  ICML 2022.

[3] CAFE: Learning to Condense Dataset by Aligning Features. CVPR 2022.

[4] Dataset Condensation with Differentiable Siamese Augmentation. ICML 2021.

[5] Dataset Condensation with Gradient Matching. ICLR 2021.

**Summary And Contributions:**

This paper proposed a benchmark for dataset decoupling task. The dataset decoupling task can separate the task-related features and sensitive attributes. The paper includes various decoupling methods and evaluates the privacy-utility tradeoff. Finally, the code and data are released.

---

> ### Author Rebuttal · Authors · 2024-08-16
>
> We appreciate our reviewer’s recognition of the novelty of our work as the first benchmark for decoupling methods. We ensure to integrate your insightful feedback into our work.
>
> **Q1: How the benchmark is set up is not clear from the paper:**
>
> A1: As demonstrated in **Figure 1**, our benchmark framework addresses and standardizes all the aspects around decoupling methods in the context of synthetic image generation. We believe with this standardization and the variety of our experimental setups, we address all the aspects of benchmarking decoupling methods and unify the common practices in one framework. In summary our benchmark setup is as follows:
> + **Data splits: line 122-135, 177-194 and Table 1** the current datasets used for decoupling provide train/val sets leading to each study defining their own splits. We standardize the data splits (train, val, generation, and test) to ensure equal attribute distribution and facilitate fair comparison across studies.
> + **Decoupling Algorithms:** we integrate 3 decoupling methods in our benchmark: CE **(line 87-92, 215-219)**, CE+masking **(line 93-97, 219-222)**, and Metric Learning **(line 98-105, 222-227)**. We use Stable Diffusion with Low Rank Adaption (LoRA) and appropriate losses to train each of these methods and save the LoRA weights for image generation **(line 196-212)**.
> + **Image Generation Pipeline: line 228-236** We load the trained LoRA weights into a standardized image generation pipeline; and use each image-prompt pair from generation data split to generate a new image conditioned on this pair. The generated images are to retain the non-sensitive attributes and to anonymize the sensitive one.
> + **Evaluation Metrics: line 136-158** We include four evaluation metrics to measure quality of generated images as well as privacy-utility trade off:
>   - **Utility FID**:  compares the distribution similarity between real and synthetic images.
>   - **Utility CLS Evaluation**: measures utility performance by quantifying the classification of non-sensitive attributes from decoupled synthetic images. (the higher the better).
>   - **Privacy CLS**: measures privacy leakage by quantifying the classification of sensitive attributes from decoupled synthetic images The accuracy should be ~random, indicating that it is not possible to recognize the sensitive attribute from the generated dataset.
>   - **Privacy Pretrained (PT)**: Using the trained base and adversary model, we run inference on real images before synthesizing to measure classification accuracy. PT quantifies privacy performance in isolation from the effect of image generation pipeline.
> + **Experiment setups: (line 239-248, Table 2-5)** In addition to the common evaluation method,  setup A, we include experiment setups B and C to evaluate cross generalization of the decoupling methods. Given that two of the privacy-utility tradeoff metrics require training a classifier, we demonstrate the effect of amount of data on these metrics **(line 256-262, section A1 supp material, Table 2-5)**. Lastly, since many studies only rely on frontal faces, we repeat all the experiments in two setups: evaluate only on frontal faces; and train and evaluate on frontal faces **(section A2.3 and A2.2 supp materials)**.
>
> **Q2:  There is no systematic evaluation of previous dataset decoupling methods. The benchmark analysis and contribution is not sufficient.**
>
> A2: as discussed in Q1, our goal is to capture different aspects where decoupling studies resort to individual assumptions and provide a unified protocol for the community to implement/evaluate their methods rather than benchmarking as many methods/datasets. We provide detailed explanations, code, and synthetic datasets and aim to publicly host a project page for the community to build upon this framework.
>
> Regarding the generalization to other tasks, from a human standpoint there are many attributes to be decoupled such as gender, age, ethnicity, hair color, emotions, etc. Once one knows which attributes to anonymize or retain, the same principle applies; with the only difference being the classification labels. In our framework, none of the proposed modules are based on a certain attribute, which makes our framework versatile and task agnostic.
>
> Given the discussion provided in Q1 and above, we respectfully disagree that our analyses and contributions are not significant. We believe this framework serves as a solid foundation for future research/development. Given this, if our reviewer has a specific aspect in mind that we did not consider, we will be glad to discuss and take that into consideration.
>
> **Q3: Literature review and future work.**
>
> A3: We tried our best to review and cite all the relevant literature on decoupling works. If our reviewer is aware of a study on attribute decoupling that we might have missed to include, we greatly appreciate it if they could kindly share it with us.
>
> Thank you for your suggestion. Please see our global response for takeaways and future work.
>
> **Q4: Dataset condensation and how it relates to the idea of decoupling when it comes to privacy.**
>
> A4: The goal of attribute decoupling techniques is to acquire a decoupled version of a private dataset **with the same distribution and same number of samples where the sensitive information is anonymized, while the non-sensitive attributes and realism of images are maximally preserved (line 71-77)**. Dataset condensation, originally, aims for efficient training by condensing a large dataset into a smaller synthetic dataset that is comparable to the original one. We acknowledge that in ref 2 the DC technique is shown to be effective for privacy preservation. However, aside from the fact that this study also uses DC for privacy, respectfully, we do not find other links between decoupling techniques and data condensation. If our review observes further links, we would appreciate it if they could kindly share their view points with us.

---

> > ### Author Response · Authors · 2024-08-23
> > **Friendly Reminder**
> >
> > Dear Reviewer NgJu,
> >
> > This is a gentle reminder that we are approaching the midpoint of the discussion period. Your feedback and suggestions have been instrumental in improving the quality of our paper, and we are sincerely grateful for your contributions.
> > We have made significant efforts to address your concerns through our responses. Specifically, we have:
> > + Clarified the contributions of our paper and elaborated on our motivation behind this paper. Additionally, we expanded on takeaways and future work of our study.
> > + Expanded on different components of our benchmark set up and further clarified each of our benchmark components including the decoupling methods.
> > + Addressed your concerns regarding literature review and the relation of our work to data condensation.
> >
> > We sincerely hope to receive your feedback and look forward to answering any additional questions you may have. Thank you once again for your valuable input.
> >
> > Best regards,
> >
> > The Authors

---

### Author Rebuttal · Authors · 2024-08-16

### **Global Response**

We want to greatly appreciate our reviewer's insightful feedback and comments which help us to improve our work. We are delighted to have received many great acknowledgements of our efforts and contributions through their comments. Reviewer **NgJu** acknowledges this is the _**“first benchmark for dataset decoupling methods”**_.

Reviewer **mYev** emphasizes on _“**novelty and comprehensiveness** of our benchmark which **fills a significant gap** in the field by providing a systematic method for evaluating privacy-preserving techniques”_. Additionally, Reviewer **mYev** acknowledge our _“**extensive experiments** and **well-documented results** which show clear improvements in privacy-utility trade-offs”_. Reviewer **mYev** believe our _“**contributions are substantial** and valuable for the research community and **can accelerate advancements in privacy-preserving machine learning**”_.

Reviewer **fx2c** emphasizes our framework is _“**versatile**, and **applicable across various domains** and data types”_ and its _“**flexibility** enhances its utility and relevance in a wide range of scenarios”_. Additionally, Reviewer fx2c mentions _“evaluating both privacy and data utility, **offers a holistic view** of the trade-offs involved, **providing a balanced assessment of decoupling techniques**”_. Reviewer **fx2c** acknowledges that our _“experimental setup is **meticulously crafted**, using detailed evaluation protocols”_  which _“**allows for a deep understanding of the performance, robustness, and generalizability** of different decoupling techniques under various conditions”_.

**Contributions:**

we introduce the first benchmark for dataset decoupling methods. Our contributions are as follows:
+  introducing a comprehensive framework that systematically addresses and standardizes all the aspects around attribute decoupling in the context of synthetic data generation, including  i) Decoupling algorithms, ii) Image generation pipeline, iii) Structured data splits, and iv) Evaluation metrics for utility and privacy.

+  integrating decoupling techniques in the context of synthetic image generation and their practical considerations.

+  releasing a dataset of images with decoupled attributes, generated using our data generation pipeline; which enables researchers to evaluate their decoupling methods within our framework, utilizing the provided metrics.

+ establishing a comprehensive benchmark by extensively evaluating several decoupling techniques for privacy-preserving image synthesis . Our benchmark systematically quantifies the privacy-utility trade-off.

**Differential Privacy Experiment:**

Per reviewer **fx2c** feedback we added differential privacy to our setup A **(Table 2)** for the CE model. We used Opacus Privacy Engine  with epsilon_tolerance=0.1, target_delta=1e-5, and epsilon 5. We report the following numbers:

| Method (V+G)| Privacy (CLS, Race) | Utility (CLS,Age)| Utility (CLS,Gender) |
| ------ | ------| ------ |------ |
| Baseline | 55.86  |  39.45 | 74.69|
| CE   | 46.23   | 34.44  | 73.13|
| CE+DP   | 39.21  | 35.04 | 73.78|

As observed in the table above, CE+DP provides better privacy while giving similar CE utility performance, and degrading wrt. Baseline which is expected. Similar to above, we can add DP for the rest of the methods across the setups.

**Computational Complexity:**

Per reviewer **mYev** feedback we expand on computational complexity of our models. As mentioned in **line 268-269** we use 1xH100 GPU.

+ **Training LoRA:** the conditional Unet number of parameters including added LoRA layers is 859.62M from which 99.840K are for trainable LoRA adapters. For CE an additional FC layer is added which adds 1.5K number of parameters (using 128x128 image resolution). In the CE+mask model the filter generation network adds 90K trainable parameters.
The overall model including image encoder (VAE) and text encoders with batch size of 128 takes 23 GB GPU memory and 2 Hours to train for 15 epochs. In this setup, the number of FLOPs are: ~4.33 TFLOPs for VAE,   ~2.89 TFLOPs for Unet backbone, and ~0.85 TFLOPs for text encoder.
+ **Image Generation:** the image generation pipeline occupies 13 GB of GPU memory and takes 4 Hours to generate images from _G_ split (~6K images) with batch size of 1, which approximately translates to ~1-2 seconds per image on H100 with ~74.68 TFLOPs.
+ **Classifiers:** we use ResNet18 (from scratch) to train privacy and utility classifiers which has ~11M parameters, 151 GFLOPs and occupy ~2.5 GB of GPU memory with batch size of 128. The training on _G_ split takes 0.5 Hours and the inference time is less than 1 minute on the test set.

**Takeaways and Future work:**

Per reviewer **NgJu** suggestion, we expand on our takeaways and future works. In our future works we aim to address the contributions below:
+ Ideally, one wants to add Differential Privacy on top of the existing algorithms in order to provide a formal guarantee for privacy. Please see above for results.
+ In our current study, we distributed and added the LoRA layers in a balanced way across the backbone. However, one can study and find the most effective layers for adding the LoRA weights considering privacy-utility trade off. Since most of the human attributes include rather high frequency features this task is not trivial and requires in depth study of internal diffusion mechanisms.
+ We aim to add stronger anonymization methods taking into consideration regulations such as the General Data Protection Regulation (GDPR) and add interpretability for privacy leakage. We aim to add attack/defense mechanisms to our framework and study the interaction of each attack mechanism with the decoupling algorithm.

---

### Author Response · Authors · 2024-08-28
**Kindly seeking support for review feedback before the discussion period ends**

Dear SAC and AC,

Thank you very much for your efforts and those of all the reviewers, as well as the insightful comments provided. The comments and suggestions  have been crucial in helping to improve the quality of our paper. We are grateful for your and the reviewers' contributions and are committed to integrate the reviewers' great comments into our work.

During the rebuttal phase, we endeavored to address all the concerns raised by the reviewers in our responses. Additionally, at the midpoint of the  Author-reviewer discussion period, we sent a gentle reminder to each  reviewer, asking if they had any further questions. Given that the end  of the Author-reviewer discussion period is arriving, we would greatly appreciate having our reviewers' final opinions on our work.

We appreciate your assistance on the matter in advance.


Best regards,

The Authors

---

### Decision · Program_Chairs · 2024-09-26

**Decision:**

Accept (Poster)

**Comment:**

This paper introduces a framework for integrating decoupling techniques into synthetic data generating, enabling privacy-preserving data release. The paper benchmarks multiple existing techniques on multiple datasets, and performs comprehensive evaluations on their utility and privacy performance. While the AC believes that, as a benchmark paper, more datasets and tasks (as claimed being task-agnostic) can be added to fully demonstrate the effectiveness of this pipeline, this paper introduces an important benchmark in systematic design of privacy-preserving data release, which can facilitate future researches in improving the utility-privacy tradeoff based on this framework.